

# Changes to the tropical circulation in the mid-Pliocene and their implications for future climate

Shawn Corvec[1] and Christopher G. Fletcher[2]

[1]Department of Applied Mathematics, University of Waterloo, 200 University Ave W, Waterloo, Ontario, Canada, N2L 3G1
[2]Department of Geography and Environmental Management, University of Waterloo, 200 University Ave W, Waterloo, Ontario, Canada, N2L 3G1

*Correspondence to:* Shawn Corvec (scorvec@uwaterloo.ca)

**Abstract.** The two components of the tropical overturning circulation, the meridional Hadley circulation (HC) and the zonal Walker circulation (WC), are key to the re-distribution of moisture, heat and mass in the atmosphere. The mid-Pliocene Warm Period (mPWP; ∼3-3.3 MY BP) is considered a useful analogue of near-term future climate change, yet changes to the tropical overturning circulations in the mPWP are poorly understood. Here, climate model simulations from the Pliocene Model Inter-

comparison Project (PlioMIP) are analyzed to show that the tropical overturning circulations in the mPWP were weaker than pre-industrial, just as they are projected to be in future climate change. The weakening HC response is consistent with future projections, and its strength is strongly related to the meridional gradient of sea surface warming between the tropical and subtropical oceans. The weakening of the WC is less robust in PlioMIP than in future projections, largely due to intermodel variations in simulated warming of the tropical Indian Ocean (TIO). When the TIO warms faster (slower) than the tropi-

cal mean, local upper tropospheric divergence increases (decreases) and the WC weakens less (more). These results provide strong evidence that changes to the tropical overturning circulation in the mPWP and future climate are primarily controlled by zonal (WC) and meridional (HC) gradients in tropical-subtropical sea surface temperatures.

## 1  Introduction

The tropical overturning circulation driven by tropical convection is an important driver of the hydrological cycle of the tropics

and subtropics. Additionally, variability in tropical convection can cause changes to mid-latitude weather patterns on a wide range of timescales (e.g., Weickmann et al., 1985; Roundy et al., 2010; Lau and Phillips, 1986; Trenberth et al., 1998). The tropical overturning circulation can be broken down into zonal (Walker circulation; henceforth WC) and meridional (Hadley circulation; henceforth HC) components. Both the WC and HC are susceptible to external forcing, and numerous studies have shown - both theoretically and through modeling - that the the WC and HC (at least in the Northern Hemisphere) will weaken in

response to climate change (e.g., Vallis et al., 2015; Bony et al., 2013; He and Soden, 2015; Vecchi and Soden, 2007; Gastineau et al., 2009). This is associated with an overall weakening of the global hydrological cycle and weaker convective updrafts over the tropical Oceans (e.g., Vecchi and Soden, 2007; Held and Soden, 2006).

Observation-based analysis for the historical period, and modeling studies for historical and future periods, however, are inconclusive over important aspects of the tropical circulation response to climate change. An alternative line of investigation



is to use a paleoclimate epoch, such as the mid-Pliocene warm period (henceforth mPWP; c. 3-3.3 MY BP), as an analogue of present and near-future climate (e.g., Robinson et al., 2008). The mPWP is characterized by relatively minor differences in topography and coastlines, orbital parameters, and solar forcing; importantly, atmospheric $CO_2$ concentrations were very similar to current levels (Sun et al., 2013; Haywood et al., 2009; Shukla et al., 2011). The advantage of past climate analogues

is the availability of paleoclimate data to validate climate simulations. Previous modeling studies examining changes to the tropical circulation in the mPWP found a similar weakening of the WC and HC as those studying future climate (Kamae et al., 2011; Sun et al., 2013), but these are single model studies only where the results could be skewed by individual model biases.

One of the key features of the mPWP derived from paleoclimate records was significantly warmer high-latitude regions compared to present, with relatively similar tropical sea surface temperatures (SSTs) to modern (Kamae et al., 2011; Dowsett

et al., 1996). As well there are strong indications that the upwelling regions off the coast of western South America were much warmer than present, causing a weakening of the west-east SST gradient in the tropical Pacific and indicating a flattening of the the thermocline (Dekens et al., 2007; Dowsett and Robinson, 2009). These changes in zonal and meridional SST gradients could have had important effects on the tropical overturning circulation of the mPWP and studying these effects could provide insight into the usefulness of the mPWP as an analogue for future climate change as they differ significantly from present day.

Multiple effects due to increased greenhouse gases (GHGs) can slow down the tropical overturning circulation (Vallis et al., 2015). Initially, GHG forcing alone warms the free troposphere, which tends to increase atmospheric static stability over the tropical oceans and reduce convective precipitation. This effect, sometimes referred to as the "direct" GHG forcing, occurs almost immediately after GHG is increased, as the mid-upper troposphere warms in response to decreased radiative cooling to space (Bony et al., 2013). An indirect effect of GHG forcing is warming of the surface, with the land warming almost

instantaneously, and ocean warming lagging by several decades, causing increased land-sea temperature contrasts (Bayr and Dommenget, 2012). This tends to lower surface pressures over land, relative to the oceans, which leads to an increased land-sea pressure gradient, moisture convergence and precipitation over land, and a strengthening of the monsoonal circulations (e.g., Hu et al., 2000). The delay in surface warming over the oceans increases static stability there as the free troposphere warms, while stability is either unchanged, or decreases, over land where surface temperatures can warm at a similar rate to the free

troposphere (Bayr and Dommenget, 2012).

Eventually the SSTs also warm, increasing boundary layer specific humidity, which—assuming fairly constant relative humidity under climate change (Vecchi and Soden, 2007)—increases non-linearly with increasing temperature by the Clausius-Clapeyron relation. In order for precipitation to balance evaporation on a global scale, and since most precipitation occurs through convection in the Tropics, the tropical convective mass flux must decrease (Held and Soden, 2006). In other words,

tropical convective updrafts do not have to be as strong to maintain the balance between evaporation and precipitation as before due to the increased water vapor content due to climate change.

A factor that can alter local (although not global mean) tropical convection from increasing GHGs is the spatial pattern of the SST warming. Regions where SSTs warm faster (slower) than the tropical mean are likely to experience smaller (larger) increases in static stability than the tropical mean. This is because the warming of the tropical troposphere is fairly uniform

as temperature gradients are difficult to maintain there (Ma and Xie, 2012). This change to the regional pattern of tropical



convection, caused by the pattern of SST warming and increased land-sea contrast, could lead to a redistribution of tropical and subtropical precipitation (e.g., Bayr and Dommenget, 2012; Chadwick et al., 2012).

The weakening of the tropical zonal overturning circulation (WC) is among the most robust responses of the tropical circulation to future climate change (He and Soden, 2015; Vecchi and Soden, 2007). The primary driver of the projected slowdown of

the WC is thought to be from warming SSTs and associated reduction in convective mass flux (Held and Soden, 2006; He and Soden, 2015; Ma et al., 2011; Lu et al., 2007; Vecchi and Soden, 2007). Additionally, it is well known that the WC strength can be modulated by the zonal gradient in tropical SST (Bjerknes, 1966); for example, on interannual timescales the warm phase of ENSO (El Niño) reduces the zonal SST gradient and weakens the WC. Thus, any change to zonal SST gradients resulting from increased GHGs is likely to be important for modulating the WC response to climate change. During the mPWP, the

zonal SST gradient in the tropical Pacific is thought to have been much weaker than present—primarily due to warmer SSTs in the upwelling regions off the west coast of South America—causing the WC to be weaker than present (Kamae et al., 2011). Observations and reanalysis data show contradictory trends in recent WC strength over the satellite era, with some indicating weakening (e.g., Vecchi et al., 2006; Power and Kociuba, 2010), and others indicating strengthening (e.g., L' Heureux et al., 2013; McGregor et al., 2014). Nonetheless, future projections from the Coupled Model Intercomparison Project Phase 5

(CMIP5) (Taylor et al., 2011) indicate a robust weakening of the WC and overall tropical overturning circulation over the 21st Century, which is associated with reductions in the zonal gradient of tropical SSTs (Shin and Sardeshmukh, 2010; Ma and Xie, 2012; He and Soden, 2015).

The weakening of the meridional overturning circulation (HC) in response to climate change is less robust than the weakening of the WC. A poleward expansion of the descending branch of the HC in the wintertime northern hemisphere is considered

the most robust projection for the future response (Lu et al., 2007; Kang and Lu, 2012). Additionally, as the tropical SSTs warm, the HC is expected to expand vertically, leading to a higher tropopause in the tropics, an effect which may already be apparent in observations (Santer et al., 2003). The CMIP5 models show fairly good agreement for a weakening of the Northern Hemisphere HC, with substantial disagreement over the response of the Southern Hemisphere cell (He and Soden, 2015; Ma and Xie, 2012; Vecchi and Soden, 2007). However, satellite observations suggest that the HC has, in fact, strengthened rather

than weakened since 1979 (Mitas and Clement, 2005; Liu et al., 2012). This apparent contradiction may be the result of natural variability; for example, internal fluctuations in the tropical and extratropical oceans, have effects on the tropical circulation in the short term that could be masking a longer term trend (Kosaka and Xie, 2013). Part of the variability in HC strength could be explained by changes in meridional SST gradients, which have been shown to weaken (strengthen) the HC if these gradients weaken (stengthen) (e.g., Seo et al., 2014; Levine and Schneider, 2010; Williamson et al., 2013; Gastineau et al., 2009; Kamae

et al., 2011). The meridional SST gradient from the tropics to poles is greatly reduced in the mPWP (e.g., Haywood et al., 2013; Dowsett et al., 1996; Dowsett and Robinson, 2009), which to leading order weakens the HC as shown in Kamae et al. (2011).

Our study aims to examine the tropical circulation of the mPWP using multiple general circulation models (GCMs). Our paper attempts to show the similarities and differences of the tropical circulation of the mid-Pliocene to the tropical circulation

under anthropogenic climate change. This analysis is hoped to be useful for any analysis that uses the mPWP as an analogue for





future climate change. The structure of this article will be as follows: Section 2 will explain the model dataset and diagnostics that will be used for our analysis in the data and methods section. Then Section 3 will examine the modeled tropical circulation of the mPWP with comparisons to CMIP5, broken down into one section for the WC and one for the HC. Section 4 will present further discussion and our primary conclusions.

## 2 Data and methods

### 2.1 Climate model simulations

To examine the tropical circulation of the mPWP, we used the Pliocene Model Intercomparison Project (PlioMIP), which aims to simulate the climate of the mPWP (Haywood et al., 2010, 2011). Two different groups of models were run; atmosphere-only General Circulation Models (AGCMs) and coupled ocean-atmosphere General Circulation Models (AOGCMs). The AGCM models have a prescribed monthly SST climatology, and do not change through the integration, while the AOGCM SSTs are allowed to evolve freely. Hereafter, we refer to the AGCM model group as "PRES" (for prescribed SSTs) and the AOGCM group as CPLD (for coupled ocean-atmosphere models). The mid-Pliocene boundary conditions are based on the Pliocene Research, Interpretation and Synoptic Mapping phase 3 (PRISM3) paleoclimate reconstructions (Dowsett and Robinson, 2009; Lunt et al., 2012; Salzmann et al., 2009). A list of the PlioMIP models examined in this study is included in Table 1, and further details on the experiments and model configuration can be found in Haywood et al. (2013).

The experimental set-up for each group of models is as follows. Atmospheric $CO_2$ levels are set at 405 ppm for both groups of models, while all other GHGs are kept the same as the pre-Industrial (PI) control runs. Additionally, the solar constant, orbital parameters and aerosol concentrations are kept the same as PI as well. Sea surface temperature, coastlines, vegetation type, and ice sheet boundary conditions are specified from PRISM3 (Haywood et al., 2010, 2011). The models in PRES are run with an integration length of 50 years, with 20 years for spin-up, giving 30 years of mid-Pliocene climatology. The models in CPLD are run with an integration length of 500 years or longer to allow the ocean to fully equilibrate to the increase in GHGs. Note that these PRISM3 boundary conditions are supposed to represent the average conditions of the warm periods over the approximately 300,000 year mid-Pliocene warm period. It has been suggested by some (e.g., Salzmann et al., 2013), that using paleoclimate reconstructions over this long of a period may not be appropriate as there were shorter term fluctuations in climate (mainly forced by orbital configuration) over this period and proxy data may not be representing conditions from the same mini "time-slice".

### 2.2 Climate diagnostics

To compute diagnostics to measure tropical circulation change, some post-processing was necessary on the data from PlioMIP. The data was interpolated to a 2.5x2.5 degree grid for all models to create a common resolution for computing multi-model mean statistics. We define the response as the difference, for any given quantity, between the mpWP simulation and the PI control simulation for each model. The response is evaluated using monthly mean climatologies computed over all available



years of simulation (typically 150 years in most models). Meridional wind data was available on 10 common vertical levels
between 1000 and 100 hPa for all models listed in Table 1. Lastly, if a model did not contribute output to the CPLD experiment,
it was not included in our analysis.

We compute multi-model means for PRES and CPLD using the models listed in Table 1. Each model's response is first
scaled by its tropical mean (30° S − 30° N) surface temperature response in CPLD (Table 1), consistent with the procedure
followed for future climate simulations by He and Soden (2015). In figures showing multi-model means, stippling indicates
regions of agreement across the model ensemble, and is added where all but one member in the model group agrees on the sign
of the response. The small number of models in each PRES and CPLD prevents a more robust statistical analysis of the model
agreement. JJA refers to a 3-month June-July-August mean, while DJF refers to a 3-month December-January-February mean.

To diagnose changes to the HC, zonally averaged meridional mass streamfunction ($\Psi$) was calculated (Oort and Yienger,
1996), then latitude-height cross-sections of this quantity were created. This field highlights the vertical and meridional circula-
tion associated with the HC, with positive (negative) values indicating clockwise (counterclockwise) overturning. To diagnose
changes to the WC, we use upper-tropospheric (200 hPa) velocity potential ($\chi$), which is anticorrelated with regions of di-
vergence (convergence) that are strong indicators of upwelling (downwelling) associated with the WC. We also use sea level

pressure (SLP) to diagnose changes to the WC and overall tropical overturning circulation (SLP data was not available from
the MIROC4m model).

## 3    Simulated changes to the tropical circulation during the mid Pliocene

### 3.1    Response of the Tropical Climate

We begin with a brief overview of the simulated changes to the multi-model mean, annual mean tropical climate in the mPWP,

relative to the pre-industrial (PI). In CPLD the tropical Oceans warm by ∼0.5-1.5K (Fig. 1a), and tropical land regions generally
warm even more than the oceans (upwards of 5K in some arid regions). The 850 hPa wind response indicates a strengthening of
the easterly trade winds across much of the tropics and subtropics, with indications of strengthened southwesterly monsoonal
flow over the northwestern Indian Ocean.

Over a large region of the tropical Ocean, the imposed SST perturbation in PRES (Fig. 1b) shows weak warming compared

to PI. The only region that sees significant warming in the tropics in PRES is the eastern Pacific, especially near the coast of
South America, with some regions warmer by up to 5 K, which drastically weakens the west-east SST gradient in the tropical
Pacific. The PRES simulations do warm in the subtropics, and show even greater warming (by a factor of 1.2 in the NH and
1.1 in the SH) than CPLD in the extratropics, especially at high latitudes (not shown). Additionally, the tropics (30° S − 30°
N) warm more in CPLD (1.75K) than in PRES (0.95). This acts to reduce the meridional equator-pole surface temperature

gradient more in PRES than in CPLD.

Tropical precipitation generally increases in CPLD, particularly over the oceans at the outer edges of the climatologically
"wet" zones (Fig. 2a). This pattern indicates an expansion of the wet regions on both sides of the equator, but with a significantly
stronger response in the northern hemisphere (NH). Land precipitation increases substantially in a band covering the Sahel,



middle East and south Asia. By contrast, precipitation decreases over the South Atlantic, the eastern South Pacific Convergence Zone (SPCZ), and over continental southern Africa and Amazonia.

The precipitation response in PRES (Fig. 2b) is rather different to that in CPLD, with moderate decreases over the climatological wet regions, and increases over much of the southern tropical ocean including over the eastern SPCZ. While precipitation

increases poleward of the northern ITCZ in CPLD and PRES, in PRES it decreases on the southern flank along the equator. Land precipitation shows similar, or even stronger, increases than CPLD in the band between north Africa and south Asia, and over Australia, regions where the simulated response agrees closely with available paleoclimate records (Salzmann et al., 2013; Kamae et al., 2011). The enhanced rainfall over relatively arid land regions is consistent with enhanced monsoonal activity (Bayr and Dommenget, 2012). We note that the largest differences in precipitation response between CPLD and PRES (Fig. 2)

occur where there are the largest differences in SST warming (Fig. 1).

## 3.2 Response of the Walker Circulation

The ascending branch of the annual mean WC is located over the Maritime Continent, where the region of strongest climatological upper-tropospheric divergence (negative 200 hPa $\chi$) indicates large-scale ascent and convective outflow (Fig. 3a). The compensating descending branch of the climatological WC is located over the East Pacific in a region of strong convergence.

The response of the WC to mPWP climate change in CPLD indicates a slight (∼3-4%) weakening of the $\chi$ field over the Maritime Continent (based on the multi-model mean of the percentage change of the absolute value of the minima in the 200 hPa $\chi$ field for each model in that region), a strengthening over the eastern Pacific, and a westward shift of the ascending branch over the western Indian Ocean (Fig. 3a). However, the weakening of the ascending branch is not robust, as indicated by a lack of model consensus on the sign of the response, with 2 models (NCAR CCSM4 and NorESM-L) showing a slight strengthening.

The WC response in PRES shows an unambiguous weakening, with decreased divergence over the western Pacific and Indian Ocean, and increased divergence over Africa (Fig. 3b). The pattern of $\chi$ response projects negatively onto the background climatological $\chi$: weakening of the divergence is focused in the regions of strongest climatological ascent, extending from the Indian Ocean into the western and central Pacific, and increased divergence is found over the eastern Pacific, near the climatological descending limb of the WC. The overall weakening response of the WC in PRES is highly robust among the

different models—more so than for CPLD—with unanimous agreement on the sign, which suggests a role for tropical SST changes. However, there is a large spread among the 4 models in PRES for the WC weakening of ∼1-29% using the same WC strength change metric used for CPLD.

Another way to examine the WC response is through the surface response, by using sea level pressure (SLP) to measure of changes in the mass distribution of the atmosphere. In Figs. 4a,b we show, for both groups of models, that the SLP response

displays a similar pattern of WC response to that obtained using $\chi$. In CPLD, the Tropics show relatively minor changes in SLP, with an increase over the Maritime Continent and western Pacific, and a decrease over the Indian Ocean associated with increased ascent and upper-level divergence there (c.f Figs. 4a,3a). This pattern represents a minor weakening of the climatological zonal gradient in tropical SLP (not shown), consistent with weakening of the WC in CPLD. By contrast, the PRES experiment shows an increase in SLP across the Maritime Continent and into the western/central tropical Pacific, with a



smaller increase in the eastern Pacific. This pattern represents a more pronounced weakening of the zonal gradient in tropical SLP, consistent with a more substantial weakening of the WC in PRES than in CPLD.

The patterns of $\chi$ and SLP response in the tropics are driven, to leading order, by changes to tropical convection which drives the mass circulation and hence the WC (e.g., Gill, 1980; Sasamori, 1982). Since the tropical troposphere warms uniformly in response to climate change, local changes in static stability must be controlled by changes in local surface temperature (Ma and Xie, 2012; Xie et al., 2010). To investigate the role of local surface warming we define the quantity dSST* as the difference between the local SST response and the tropical (20° S – 20° N) mean SST response in each model, for both PRES and CPLD (Figs. 4a,b, shading). There is a very strong association between the spatial pattern of dSST* and that of the SLP response (anomaly correlation of -0.65 for the multi-model mean and a range of -0.56 to -0.66 for the 10th-90th percentile of CPLD models), and also with 200 hPa $\chi$ (anomaly correlation of -0.57 for the multi-model mean, with a range of -0.43 to -0.6 for the 10th-90th percentile of CPLD models). The SLP ($\chi$) response tends to show high pressure (upper-tropospheric convergence) over regions where dSST* is negative; i.e., there is large-scale descent where the local warming is less than the tropical mean warming. This may explain why the WC weakens less in CPLD than in PRES, because the pattern of surface warming in CPLD produces a maximum located in regions of climatological divergence/ascent, particularly in the eastern Indian Ocean and around the Maritime Continent.

To further investigate the relation between the pattern of SST warming in CPLD and the WC, a correlation between dSST* averaged over 90° E – 150° E, 20° S – 20 ° N and WC strength change (using the method defined earlier in this section using 200 hPa $\chi$) was performed. A positive correlation of 0.61 was found among the models (i.e. higher dSST* tends to have less WC weakening). This shows that the rate of SST warming in and around the Maritime Continent (eastern Indian Ocean and western Pacific) is important for modulating the change in WC strength due to climate change (i.e., faster warming may lead to less weakening or in fact strengthening).

### 3.3 Response of the Hadley Circulation

In CPLD the annual mean climatology of mass streamfunction ($\Psi$) consists of two overturning cells on either side of the equator, with the southern cell being slightly stronger (Fig. 5a). These cells describe the HC: ascent occurs in the equatorial regions, with descent generally occurring in the subtropics. The response in CPLD is largely negative, and centered just north the equator but encompassing the ascending region of both cells. This implies a weakening of the ascending branch in the northern hemisphere (NH), and a strengthening in the southern hemisphere (SH). This general pattern of HC strength change is consistent with the expected changes due to future climate change as simulated by the CMIP5 coupled models (e.g., Vallis et al., 2015; Bony et al., 2013; He and Soden, 2015; Vecchi and Soden, 2007; Gastineau et al., 2009). There is also strengthening apparent of the descending branch of the HC in the NH, but model agreement is low.

The HC response in CPLD displays a distinct seasonal cycle, because the appearance of two distinct cells is an artifact of taking the annual mean. In DJF and JJA there exists, in fact, one dominant cell straddling the equator, ascending in the summer hemisphere and descending in the winter hemisphere (Figs. 5c,e). Henceforth, we refer to the seasonal cells with respect to the hemisphere containing the descending limb. The weakening of the DJF (i.e., NH) cell is very evident in CPLD, with a



tendency for greater weakening in the ascending limb (Fig. 5c). One feature that was not apparent in the annual mean is the increase in cell height, as evidenced by the positive Ψ response around 150 hPa. This is associated with a well-documented increase in tropopause height consistent with warming tropical SSTs, which through moist adiabatic adjustment, leads to a higher tropopause (e.g., Schneider et al., 2010; Santer et al., 2003; He and Soden, 2015). There is a clear poleward shift of

the descending branch of the DJF cell in the mid-lower troposphere, which would imply a poleward shift of the midlatitude eddy-driven jet.

The response of the JJA (i.e., SH) cell also shows weakening that is stronger in the ascending than descending region (Fig. 5e). However, in contrast to DJF, the weakening shows little model agreement (indicated by the lack of stippling), and indeed the JJA cell actually strengthens in 2/6 models (Fig. 6). This increased uncertainty in the response of the southern HC

has been documented for future climate change (He and Soden, 2015; Ma and Xie, 2012; Vecchi and Soden, 2007), but to our knowledge this is the first time a similar result has been shown for the mpWP. The increase in tropopause height is stronger in JJA than DJF, and there is a small—but robust—expansion poleward of the descending branch around 30°N that mirrors the result in the NH for DJF (Figs. 5c,e). Given the apparent non-robustness of the HC weakening response, the robustness of the poleward shift suggests that the strength change and meridional expansion are controlled by different processes. Interestingly,

the spatial pattern of the weakening of the JJA cell, and its nonrobustness, are very similar to the response seen among the CMIP5 models for future climate change (e.g., Seo et al., 2014). This has been linked to the slower rate of tropical warming in the SH compared to the NH, (e.g., He and Soden, 2015; Ma and Xie, 2012; Vecchi and Soden, 2007; Kang and Lu, 2012; Seo et al., 2014).

Overall, the response of the annual mean HC is quite different in PRES than in CPLD. In PRES both cells weaken, with

the largest weakening in the ascending branch of the southern cell just south of the equator (Fig. 5b). The weakening of the northern HC occurs in the center of the cell, indicating an overall weakening of cell strength, as opposed to a weakening of the ascending branch in CPLD. The southern HC, however, weakens mainly in the ascending region, which is opposite to the strengthening seen there in CPLD. There is a slight poleward expansion of the descending region of the northern HC in PRES, which is more robust in terms of model agreement than in CPLD.

Similar amounts of HC weakening are found in DJF and JJA in PRES (Figs. 5d,f). However, there are differences in the response around the edges of the cell in each season. There is strong model agreement that in DJF the ascending portion of the cell moves slightly poleward (Fig. 5d), while in JJA both the ascending and descending portions of the cell expand poleward (Fig. 5f). This robust weakening and expansion of the southern HC in the ascending region is consistent with an expansion of the tropical wet zones described in Section 3.1 (Fig. 2b), and with the single AGCM study of Kamae et al. (2011). While this

expansion of the HC is also somewhat evident in CPLD, it is much weaker and only robust for the northern cell (i.e., DJF) in the lower troposphere. An increase in tropopause height is not evident in any season for PRES, which is likely related to the lack of tropical warming in the imposed SSTs in PRES (Fig. 1b). Although PlioMIP temperature data were not archived from model layers in the mid-upper troposphere, we speculate that the warming in that region in PRES would be substantially weaker than in CPLD, because of the reduced latent heating driven by muted tropical surface warming (Seo et al., 2014; Santer

et al., 2003).



### 3.4 Causes of seasonal asymmetry in HC change

We next attempt to provide an explanation for why the largest non-robustness in the HC response—and the largest differences between PRES and CPLD—occur in JJA. Meridional gradients in SST and lower-tropospheric temperature between the Tropics and subtropics have been shown to affect HC strength in both mpWP and future climates (e.g., Seo et al., 2014; Levine and

Schneider, 2010; Williamson et al., 2013; Gastineau et al., 2009; Kamae et al., 2011). We further investigate this mechanism using the quantity dTs, which we define as the change in the meridional surface temperature gradient between the equatorial region ($5°$ S – $5°$ N) and the northern ($15°$ N – $25°$ N) or southern ($15°$ S – $25°$ S) subtropics. We compute dTs separately for the two winter hemispheres in each model, then compare the values of dTs with a common measure of HC strength in each model: the absolute change in the maximum (minimum) value of $\Psi$ ($\Delta|\Psi|_{max}$) for the northern (southern) hemisphere,

respectively (e.g., Kang et al., 2013).

Figure 6 shows that, to leading order, the seasonal asymmetry in the multi-model mean HC response in CPLD is explained by differences in dTs. The relatively small change in HC strength during JJA ($\Delta|\Psi|_{max}$ of -0.01 $10^{-11}$ kg s$^{-1}$ K$^{-1}$ in the multi-model mean) is accompanied by a small dTs in that season (0.03 K K$^{-1}$), compared to the larger values seen in DJF ($\Delta|\Psi|_{max}$ of -0.13 $10^{-11}$ kg s$^{-1}$ K$^{-1}$, 0.33 K K$^{-1}$). In other words, the stronger weakening of the NH cell than the SH cell in CPLD

could be explained by the subtropics warming faster than the tropics—and the associated weakening of the tropical-subtropical surface temperature gradient—in DJF, but not in JJA (Fig. 6). This is reminiscent of the result from Seo et al. (2014), who found a significant positive linear relationship between $\Delta|\Psi|_{max}$ and dTs among a suite of 30 CMIP5 models. However, in our sample of 6 CPLD models from PlioMIP, we find only a modest relationship ($r = -0.46$) between dTs and $\Delta|\Psi|_{max}$, and this only emerges when DJF and JJA data are pooled (not shown). The correlation values for DJF and JJA separately are not

significant and show the opposite relationship ($r = 0.39$ and $r = 0.14$, respectively).

The reason for the much weaker dTs in the southern Hemisphere in CPLD is that the southern tropical oceans warm more slowly than the northern tropical oceans (shown in Fig. 4a for the annual mean). This explains the hemispheric asymmetry in dTs, and this pattern is very similar for DJF and JJA (not shown). Therefore, we propose that the seasonal asymmetry in HC response in CPLD is, at least partially, explained by a corresponding asymmetry in the tropical SST warming between the

hemispheres. A further demonstration of this idea comes from PRES, where dTs and $\Delta|\Psi|_{max}$ are very similar in both seasons and hemispheres (Figs. 4b, 6). We note that the hemispheric asymmetry in surface warming found in CPLD, and the associated changes in the HC, are similar to those projected for future climate change from CMIP5 (e.g., He and Soden 2015).

### 4 Discussion and Conclusion

The tropical circulation response in two groups of models in PlioMIP show an overall slowdown of the zonal and meridional

components of the tropical overturning circulation. However, we found that there are significant differences in the Walker circulation (WC) and Hadley circulation (HC) response between the two model groups. The ascending branch of the WC weakens, but does so much less in CPLD than in PRES (and less than reported in CMIP5 models by He and Soden (2015)). Across the two experiments we find that the sign and magnitude of the WC response is strongly related to pattern of SST



warming, particularly around the Maritime Continent and eastern Indian Ocean. We also find that the change in the winter hemisphere meridional SST gradient between equatorial regions and the subtropics effects the HC response. We find that the southern subtropics warm more slowly than the northern subtropics in CPLD and there is thus an asymmetry in the HC response between JJA and DJF.

The dependence of the WC response on the pattern of SST warming is contrary to studies of future climate, where the weakening of the WC is found to be a robust response to global warming, and is not as strongly related to the spatial pattern of SST warming (e.g., Ma and Xie, 2012). However, there exists a two-way coupling between the WC and SSTs through the Bjerknes feedback (Bjerknes, 1966), making it possible that the models in CPLD exhibit greater Indian Ocean warming precisely because the WC weakens less. In PRES, where SSTs are held fixed, we see a significant weakening of the WC

compared to PI, which indicates that the atmospheric changes from global warming weaken the WC. This provides further evidence that the pattern of SST warming in CPLD is working against the overall tendency for WC weakening driven by an increasing atmospheric stability. Only a new set of experiments designed to separate the SST and WC changes, beyond the scope of this study, can fully resolve this problem of cause and effect.

What we have found in PlioMIP for the HC is generally consistent with projections of future climate. Thus, the PlioMIP

models are indicating that the Pliocene tropical circulation could be similar to near-future (i.e., this century) HC changes caused by anthropogenic climate change. The HC response of both groups of models is similar to projections of future climate, with CPLD having the most similar response to CMIP5. In fact, the spatial pattern of the CPLD JJA HC response looks incredibly similar to that of the RCP8.5 CMIP5 response in Fig. 1b in Seo et al. (2014). As we have shown, this similarity is likely explained by the similar meridional asymmetry in tropical SST response in CPLD and CMIP5, with regions in the south

tropical Pacific warming slower than those along and north of the equator. It would seem that the meridional pattern of tropical SST response is a robust response to increased GHGs as it is seen in both the PlioMIP coupled simulations and in CMIP5.

The spatial pattern of tropical circulation response in PlioMIP shows that regions with greater warming of the tropical SSTs tend to see increases in precipitation and upper-level divergence; i.e., the "warmer-get-wetter" pattern discussed in Ma and Xie (2012). Our results are also in agreement with the single AGCM mid-Pliocene experiments of Kamae et al. (2011) using

prescribed SSTs, and Sun et al. (2013) using an interactive ocean model, providing support that the details are robust to model configuration. However, it is striking that the pattern of tropical SST warming produced in CPLD is largely inconsistent with the PRISM3 paleoclimate SST reconstructions used as boundary conditions in PRES (Dowsett et al., 2013; Salzmann et al., 2013; Hill, 2015). We have shown that this has important implications for the tropical circulation, and so this requires urgent further investigation if the mPWP is to be considered as a reliable analogue for future climate.

*Acknowledgements.* We thank the modeling groups in Table 1 who contributed data to the PlioMIP project. SC and CF acknowledge funding from NSERC Discovery Grant 402661.



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



**Table 1.** The list of models available from PlioMIP for the prescribed SST (PRES) and coupled ocean-atmosphere (CPLD) experiments. A bullet (dash) indicates that the model data are available (unavailable). Two models in CPLD (GISS and NCAR CCSM4) do not have equivalents in PRES (we elected not to use data from NCAR CAM3.1, because it was not available on pressure levels). In the left column, the letter in parentheses after the model name shows its index in Fig. 6. The right column shows the tropical ($30°$ S – $30°$ N) mean near-surface air temperature response ($\Delta$Ts, units K) for each model in CPLD.

| PlioMIP model list | | | |
|---|---|---|---|
| Model (Letter) | PRES | CPLD | CPLD $\Delta$Ts (K) |
| GISS (A) | – | ● | 1.36 |
| HadCM3 (B) | ● | ● | 2.56 |
| MIROC4m (C) | ● | ● | 2.58 |
| MRI-CGCM2.3 (D) | ● | ● | 1.48 |
| NCAR CCSM4 (E) | – | ● | 1.24 |
| NorESM-L (F) | ● | ● | 1.89 |





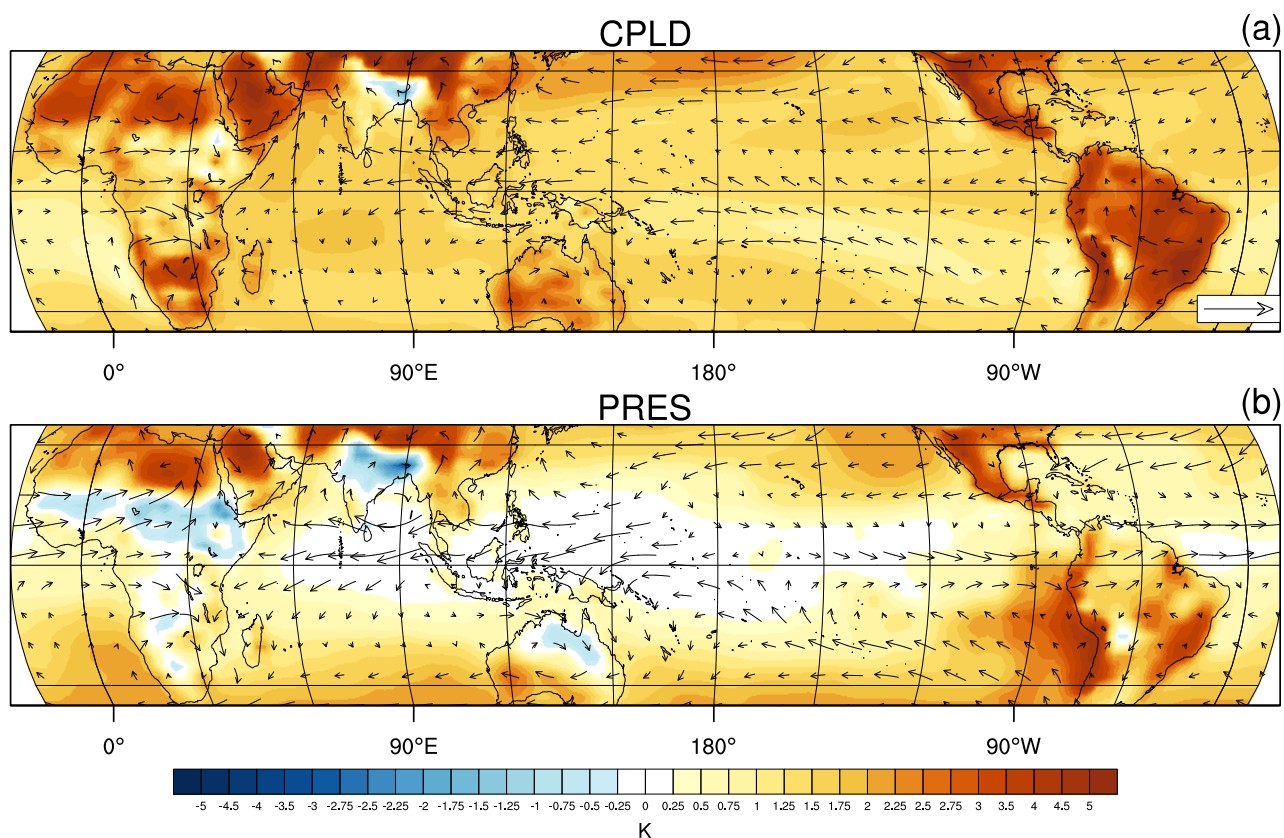

**Figure 1.** CPLD (**a**) and PRES (**b**) multi-model mean, annual mean surface temperature response (shading, units K) and 850 hPa wind response (reference vector denotes 3 m s$^{-1}$ K$^{-1}$). Prior to computing the multi-model mean, the response in each model is scaled by its tropical mean surface temperature response in CPLD (see Section 2.2 and Table 1).




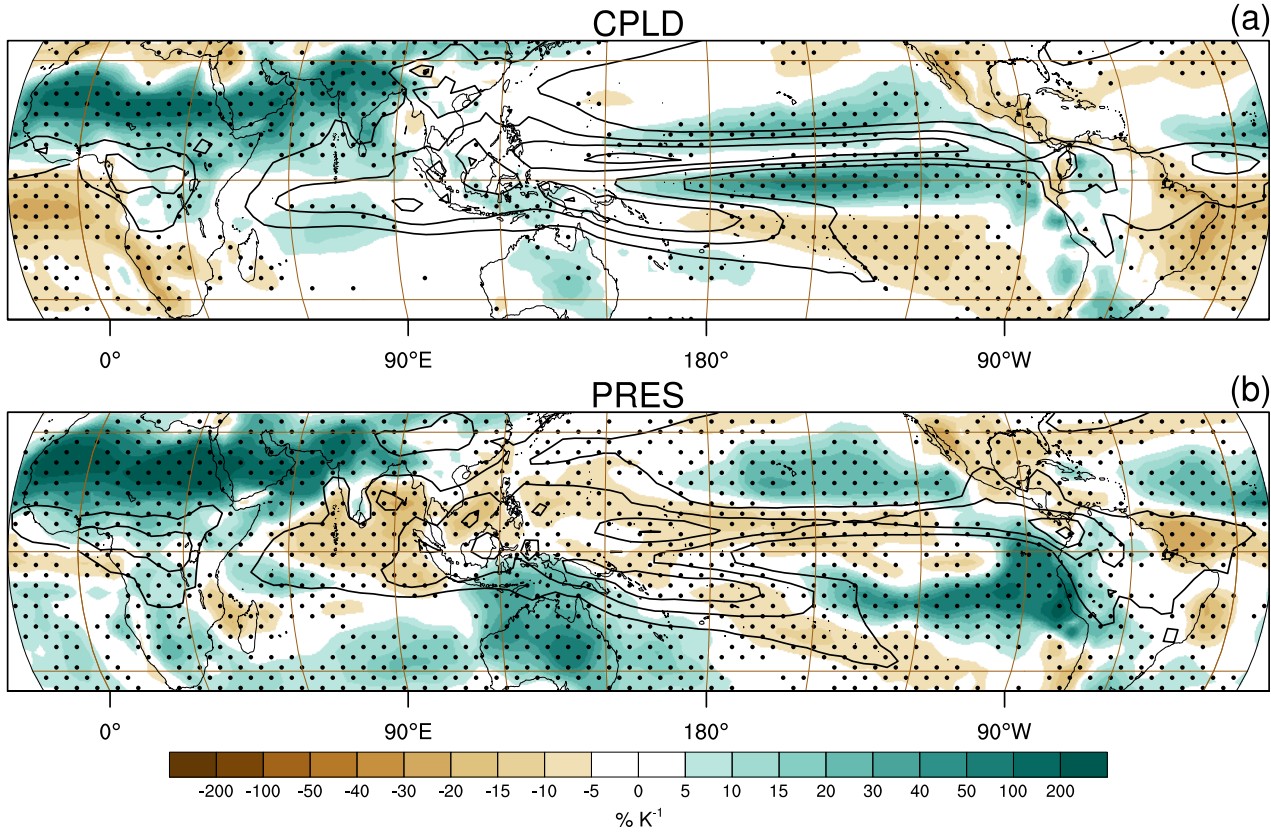

**Figure 2.** As Fig.1, except showing the precipitation response as a percentage of the pre-industrial control climatology (shading). The precipitation rate from the pre-industrial control climatology is shown by thick black contours, interval 2 mm day$^{-1}$ and starting at 4 mm day$^{-1}$. Stippling indicates regions where all but one of the models in the model group agree on the sign of the response.




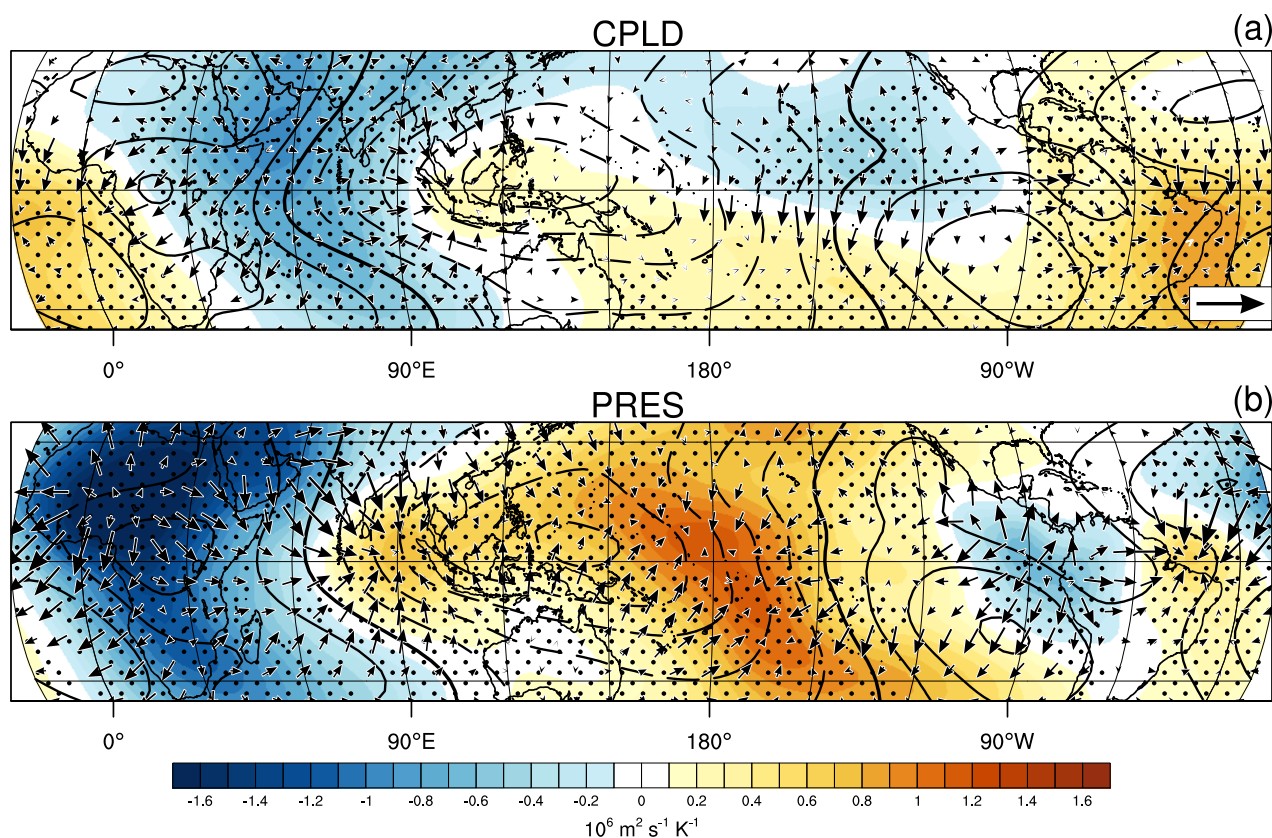

**Figure 3.** As Fig.1, except showing the 200 hPa annual mean velocity potential ($\chi$) response (shading) and 200 hPa divergent wind annual mean response vectors (reference vector denotes 1 m s$^{-1}$ K$^{-1}$). Contours show 200 hPa $\chi$ from the pre-industrial control climatology, interval 2 x 10$^6$ m$^2$ s$^{-1}$ with dashed contours negative and the zero line thickened. Stippling as in Fig.2.



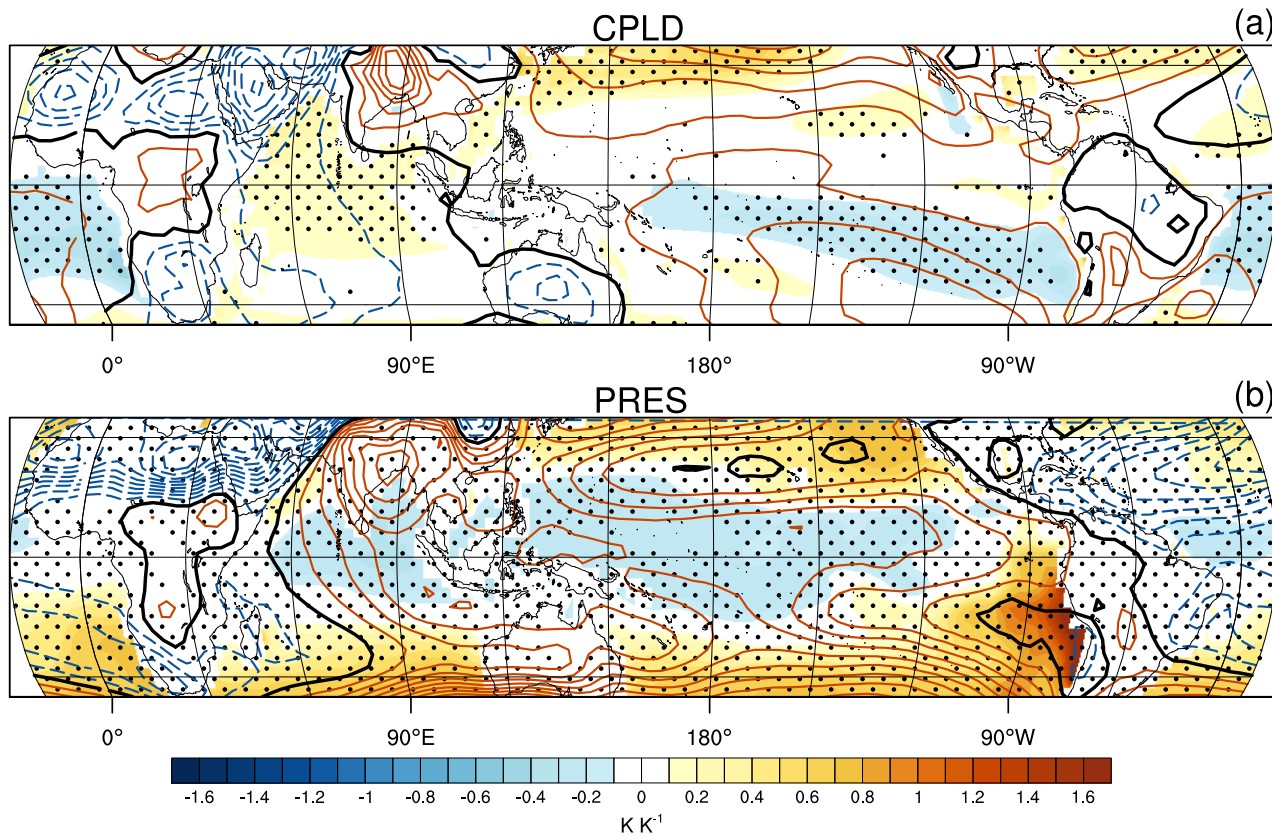

**Figure 4.** As Fig.1, except showing the dSST* response (shading) and SLP response (contours, interval 0.2 hPa K$^{-1}$ with blue negative, red positive, and black denoting the zero line). Stippling as in Fig.2.



**Figure 5.** CPLD (**a, c, e**) and PRES (**b, d, f**) meridional mass streamfunction response (shading) and pre-industrial control climatology (contours, interval 2 x $10^{10}$ kg s$^{-1}$, with dashed negative and zero line thickened): (**a**) & (**b**):, annual mean (**c**) & (**d**): DJF mean, (**e**) & (**f**): JJA mean. Stippling as in Fig.2.



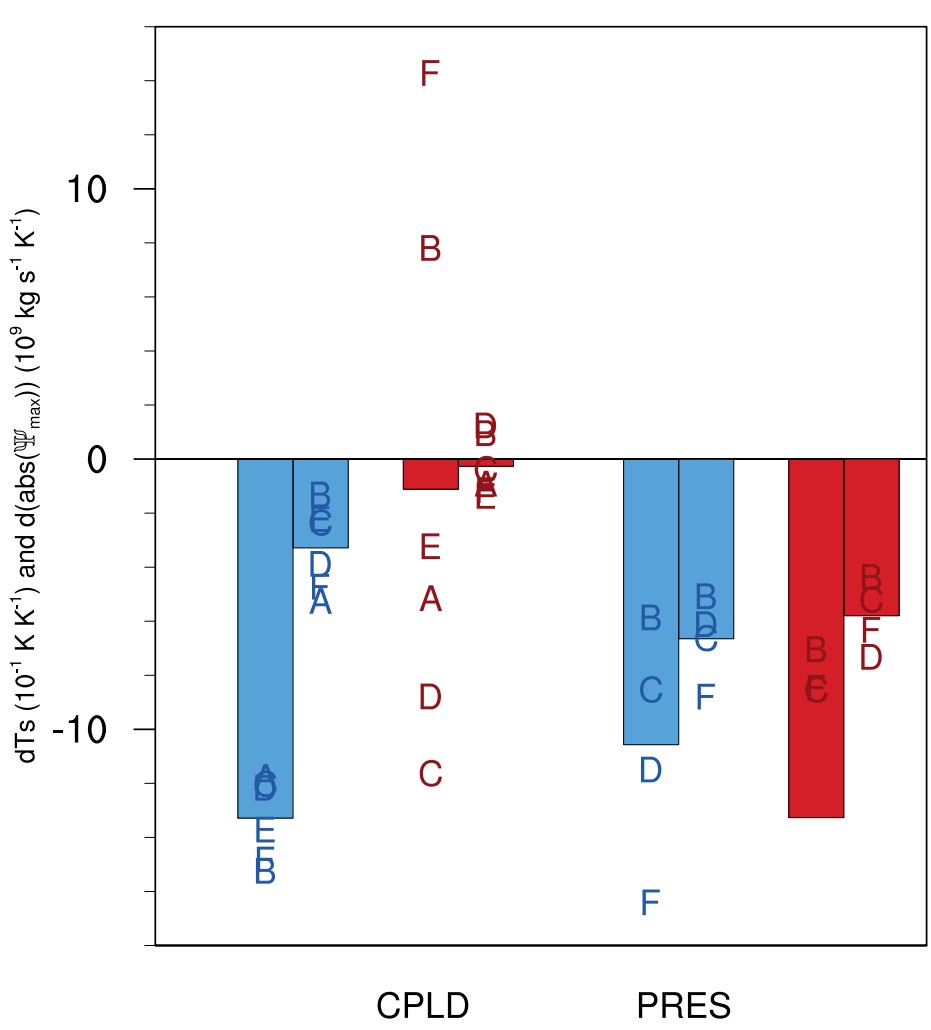

**Figure 6.** CPLD and PRES multi-model mean DJF (blue) and JJA (red) $\Delta|\Psi|_{max}$ (left bar of pair) and dTs (right bar of pair), with individual models indicated by letters (see Table 1 and text for details).