# Peer review of "Changes to the tropical circulation in the mid-Pliocene and their implications for future climate"

_Climate of the Past, 2016_

## Referee Comment (RC1) · Anonymous Referee #1 · 13 Nov 2016

The authors present an analysis of changes in tropical overturning circulation from the PlioMIP1 simulations to show both Walker circulation and Hadley circulation were weaker during the mid Piacenzian relative to the preindustrial. This has relevance for projections of future climate.

Since my expertise is not in line with the primary discussion section of this paper, my comments are mostly confined to the data cited and general aspects of the paper. While it is good to see the PlioMIP products being used to better understand Pliocene climate and inform us about future climate, there appears to be a poor familiarity with Pliocene literature. There are a number of citations that are poor choices and others simply are incorrect. The lack of familiarity with the relevant data may or may not affect

the analyses presented, but it makes for difficult reading. While the authors make use of the PlioMIP simulations and the PRISM3 boundary conditions, I do feel there are many other references outside of the PRISM group, that could be used to help support the authors arguments.

I hope the editors have chosen other reviewers better suited to comment on the analyses themselves.

Specific corrections/suggestions:

Page1/line3: stratigraphic convention is oldest to youngest; reverse your ages and use "Ma" not MY BP.

Page1/line3: I don't think you will find anyone calling it an "analogue" anymore. Several papers have been written on the topic. Maybe "scenario" or "imperfect analogue" would be a better choice.

Page2/line1: Don't use "epoch" since it is a formal stratigraphic term which would refer to the Pliocene as a whole i.e. Pliocene Epoch. Maybe "interval" would work better?

Page2/line1: 3.3 to 3.0 MA, NOT 3-3.3 MY BP; Again, "Analogue" needs to be replaced.

Page 2/line2: "relatively minor difference" | relative to what? What is considered minor?

Page 2/line4: I don't think any of the three references you cite are adequate for the first half of the sentence.

Page2/lines4-5: Even if you want to call a past interval of time an "analogue" to future climate, just having that interval does not guarantee the availability of data. The availability of data depends upon a number of factors. I think you are trying to say something along the lines of 'having a wealth of paleoclimate data available for an interval not unlike what has been projected for the future is useful for validation of climate simulations." Or something along those lines.

Page2/line9: The way this is written it sounds as though Kamae et al. produced paleo-climate records, and there are a number of other citations, in addition to Dowsett et al. (1996), that could/should be noted.

Page 4/line13: You need to cite the actual PRISM3 reconstruction which appeared in the journal Stratigraphy in 2010. These were the boundary conditions used. Neither Lunt et al. 2012 nor Dowsett & Robinson (2009) are citations for PRISM3.

Page4/line19: Again, as written it appears you are citing Haywood et al. (2010, 2011) for PRISM3. These might be good citations for the PlioMIP experimental setup but not for the PRISM3 boundary conditions themselves.

Page4/lines22-23: It would appear you need a reference here for "warm peak averaging."

Page4/line26: delete "mini"

Page10/line27: How do the CPLD and PRISM3 SST's differ? Could you show a $\Delta$SST map. How about the actual data points PRISM3 used and not the SST field which is highly interpreted and certainly not that accurate? This would be an interesting addition.

Page10/lines27-28: these citations are not correct for the PRISM3 boundary conditions (specifically the SST's) that were used for experiment 1 of PlioMIP phase 1. Maybe you mean a different Dowsett et al. (2013)? Either Dowsett et al. (2009) or Dowsett et al. (2010) would be appropriate for the PRISM3 ocean.

---

## Referee Comment (RC2) · Anonymous Referee #2 · 20 Dec 2016

This is an interesting paper that provides quite a detailed analysis of changes in tropical circulation in response to Pliocene boundary conditions in climate models. The nature of atmospheric circulation in the tropics provided by models is not new (and the authors could do with referring to additional previous studies that have demonstrated similar behavior). However, to my knowledge this is the first study that examines these aspects of model responses across an internally consistent multi-model ensemble. To me this is where the novelty of the paper resides. The paper presents analyses using both prescribed SST and predicted SST simulations from PlioMIP Phase 1, which again have been presented before for individual models but never synthesized as an MME. Finally the paper draws some interesting comparisons between model responses in

the Pliocene versus future climate change experiments, which I found very interesting and that directly face the Pliocene4Future agenda.

As a climate modeller I do not have any concerns with the methods used to analyse the model data and the approach towards the analysis of statistical robustness seems sensible to me. I think the conclusions drawn in the paper are well justified by the results presented and compliment previous work very nicely.

This paper makes an important and useful contribution to the PlioMIP project.

I recommend acceptance after only minor revisions. I formed my opinion before reading the comments of reviewer number 1 but on the whole support a number of his/her assertions regarding the lack of appropriate citation to previous mPWP studies.

Rather then analogue I would prefer the time period referenced as a unique opportunity to better understand climate dynamics and behavior in a warmer world. The implications of this are obvious without having to engage in any complex discussion on what constitutes an analogue or not.

I think in the introduction the authors should expand upon the aspects of tropical circulation response that models, when simulating future climate, are not consistent about.

Please use mPWP throughout and not mpWP.

I would also note that differences in orbital parameters in the mPWP overall were not "minor" from modern as the authors suggest. The Laskar orbital solution shows that across the ∼300 Kyrs of the mPWP that there were very large changes in insolation at the TOA. Convention in previous model simulations for the mPWP was to use a modern orbit even those the evaluation data (SSTs and vegetation) would reflect a complex response to an amalgam of orbital forcing (due to its time average nature). This is why in PlioMIP Phase 2 they are focusing on a narrower time slice ∼3.2 Ma where the orbital forcing represented by the SST responses, and given to the models themselves, is consistent.

The authors should read the Haywood et al. 2016 review in Nature Communications to familiarize themselves with current uncertainties regarding tropical SST response. For those in the community who are in the know there is currently significant debate about stability of tropical SSTs during the Pliocene. Basically one would expect a model to increase tropical SSTs in response to a CO2 increase (as indicated by CO2 proxies). But the change derived from a ~120ppmv CO2 forcing is small and inherent uncertainties in the sensitivity of proxy detection and attribution remain - meaning quite simply the signal to noise ratio in this regard is unfavorable. So it is of no surprise that the PRES and CPLD results differ and the sensitivity of SSTs in the tropics during the mPWP is an area where current research is ongoing.

---

## Author Comment (AC1) · 6 Jan 2017

article

**General comments:**

We would like to thank the reviewer for the comments and suggestions, especially with regards to suggestions for the appropriate citations of the Pliocene literature.

**Responses to each comment from reviewer 1:**

A number of citations have been added or changed. We have attempted to address each point made by the reviewer below (comments highlighted in blue, our responses in red):

**Comment 1**: *Page1/line3: stratigraphic convention is oldest to youngest; reverse your ages and use "Ma" not MY BP*:

Changed to "Ma".

**Comment 2**: *Page1/line3: I don't think you will find anyone calling it an "analogue" anymore. Several papers have been written on the topic. Maybe "scenario" or "imperfect analogue" would be a better choice*

Changed to "imperfect analogue".

**Comment 3**: *Page2/line1: Don't use "epoch" since it is a formal stratigraphic term which would refer to the Pliocene as a whole i.e. Pliocene Epoch. Maybe "interval" would work better?*

Changed to "interval".

**Comment 4**: *Page2/line1: 3.3 to 3.0 MA, NOT 3-3.3 MY BP; Again, "Analogue" needs to be replaced.*

Changed to "Ma" and "imperfect analogue".

**Comment 5, 6, 7**: *Page 2/line2: "relatively minor difference" | relative to what? What is considered minor?*

*Page 2/line4: I don't think any of the three references you cite are adequate for*

[Figure]

*the first half of the sentence.*

*Page2/lines4-5: Even if you want to call a past interval of time an "analogue" to future climate, just having that interval does not guarantee the availability of data. The availability of data depends upon a number of factors. I think you are trying to say something along the lines of 'having a wealth of paleoclimate data available for an interval not unlike what has been projected for the future is useful for validation of climate simulations." Or something along those lines.*

Rewritten as: "...The mPWP was the most recent period where $CO_2$ concentrations were similar to those levels that are projected to be reached early this century (Raymo et al., 1996) and the continental configuration was highly similar (relative to other paleoclimate intervals further in the past) to present (Dowsett, 2007a). Since a wealth of paleoclimate data is available for the mPWP, and its climate conditions are expected to be similar to the climate of the near-future (Dowsett, 2007a), this interval is considered useful for validation of future climate simulations."

**Comment 8**: *Page2/line9: The way this is written it sounds as though Kamae et al. produced paleoclimate records, and there are a number of other citations, in addition to Dowsett et al. (1996), that could/should be noted.*

Kamae et al. reference removed, and changed citations to: (Dowsett et al., 1992, 1994, 1996; Raymo et al., 1996).

**Comment 9**: *Page 4/line13: You need to cite the actual PRISM3 reconstruction which appeared in the journal Stratigraphy in 2010. These were the boundary conditions used. Neither Lunt et al. 2012 nor Dowsett & Robinson (2009) are citations for PRISM3.*

[Figure]

This citation has been changed to Dowsett et al, 2010 (The PRISM (Pliocene palaeoclimate) reconstruction: time for a paradigm shift).

**Comment 10**: *Page4/line19: Again, as written it appears you are citing Haywood et al. (2010, 2011) for PRISM3. These might be good citations for the PlioMIP experimental setup but not for the PRISM3 boundary conditions themselves.*

Changed citation to same as above

**Comment 10**: *Page4/lines22-23: It would appear you need a reference here for "warm peak averaging."*

The lines now read: "Note that the PRISM3 boundary conditions employ the technique described as "warm peak averaging" by Dowsett and Poore (1991) to represent the average conditions of the warm periods over the approximately 300 Kyr mPWP.".

**Comment 11**: *Page4/line26: delete "mini"*

Page4/line26: deleted "mini".

**Comment 12**: *Page10/line27: How do the CPLD and PRISM3 SST's differ? Could you show a $\triangle$SST map.*

Figure of ANN PRISM3 SST minus CPLD MMM SST included and was added as a third panel to Fig. 1.

**Comment 13**: *How about the actual data points PRISM3 used and not the SST field which is highly interpreted and certainly not that accurate? This would be an*

*interesting addition*

Sentence has been re-written as: "However, it is striking that the pattern of tropical SST warming produced in CPLD is largely inconsistent with the PRISM3 paleoclimate SST reconstructions (Dowsett et al., 2010, 2009) used as boundary conditions in PRES."

---

## Author Comment (AC2) · 6 Jan 2017

article **General comments:**

We would like to thank reviewer 2 for the feedback and constructive criticism. Below we hope to address the comments made with the specific comment highlighted in blue and our response below in red:

**Responses to comments from reviewer 2:**

*Rather then analogue I would prefer the time period referenced as a unique op-*

*portunity to better understand climate dynamics and behavior in a warmer world. The implications of this are obvious without having to engage in any complex discussion on what constitutes an analogue or not.*

We have changed the sentence on page 3/line 35 to read: "As such, the mPWP provides a unique opportunity to improve our understanding of large scale climate dynamics in a warmer world.".

*Please use mPWP throughout and not mpWP*

We have changed all abbrevations of mid-Pliocene Warm Period to "mPWP".

*I would also note that differences in orbital parameters in the mPWP overall were not "minor" from modern as the authors suggest. The Laskar orbital solution shows that across the âĹij300 Kyrs of the mPWP that there were very large changes in insolation at the TOA. Convention in previous model simulations for the mPWP was to use a modern orbit even those the evaluation data (SSTs and vegetation) would reflect a complex response to an amalgam of orbital forcing (due to its time average nature). This is why in PlioMIP Phase 2 they are focusing on a narrower time slice âĹij3.2 Ma where the orbital forcing represented by the SST responses, and given to the models themselves, is consistent.*

End of second paragraph in "Data and Methods" section on page 4 re-written as:

"It has been suggested that using paleoclimate reconstructions over such a long period may not be appropriate, as shorter-term fluctuations in climate forced by changes in orbital parameters mean that proxy data may not be representative of conditions from the time period consistent with the boundary conditions (e.g., Salzmann et

al., 2013). Indeed, over the ∼300 Kyr mPWP, it has been shown that solar insolation exhibited very large fluctuations (Laskar et al., 2004). The next phase of the PlioMIP project, PlioMIP2, will mitigate against this issue by focusing on a narrower time period at 3.2 Ma, with consistent orbital and SST forcing."

*"... in the introduction the authors should expand upon the aspects of tropical circulation response that models, when simulating future climate, are not consistent about."*

The second last paragraph of the introduction as follows (split into 2 paragraphs) as: "The weakening of the meridional overturning circulation (HC) in response to climate change is less robust than the weakening of the WC. A poleward expansion of the descending branch of the HC in the wintertime northern hemisphere is considered the most robust projection for the future response, although the physical mechanisms for this are not well understood (Lu et al., 2007; Kang and Lu, 2012). Additionally, as the troposphere warms, the HC is expected to expand vertically as the tropopause height in the tropics increases, an effect which may already be apparent in observations (Santer et al., 2003). The CMIP5 models show fairly good agreement for a weakening of the Northern Hemisphere HC, with substantial disagreement over the sign of the response of the Southern Hemisphere cell (He and Soden, 2015; Ma and Xie, 2012; Vecchi and Soden, 2007). However, satellite observations and reanalysis data suggests that the HC has, in fact, strengthened rather than weakened since 1979 (Mitas and Clement, 2005; Liu et al., 2012). This apparent contradiction may be the result of poor model parameterization of clouds or convection (e.g., Mitas and Clement, 2006; Sohn et al., 2016) or due to natural variability. For example, internal fluctuations in the tropical and extratropical oceans, have effects on the tropical circulation in the short term that could be masking a longer term trend (Kosaka and Xie, 2013).

Part of the variability in HC strength could be explained by changes in meridional SST gradients, which have been shown to weaken (strengthen) the HC if these gradients weaken (strengthen) (e.g., Seo et al., 2014; Levine and Schneider, 2010; Williamson et al., 2013; Gastineau et al., 2009; Kamae et al., 2011). The meridional SST gradient from the tropics to mid-latitudes is greatly reduced in the mPWP (e.g., Haywood et al., 2013; Dowsett et al., 1996; Dowsett and Robinson, 2009), which to leading order weakens the HC as shown in Kamae et al. (2011)." This reduced meridional SST gradient world can provide a test-bed for the climate models to better gauge the sensitivity of the HC to these boundary conditions.